# Magnetic Resonance Imaging and Manganism: A Narrative Review and Laboratory Recommendations

**DOI:** 10.3390/jcm13102823

**Published:** 2024-05-10

**Authors:** Michal Majewski, Karolina Piwko, Michal Ordak, Elzbieta Muszynska, Tadeusz Nasierowski, Magdalena Bujalska-Zadrozny

**Affiliations:** 1Department of Pharmacotherapy and Pharmaceutical Care, Faculty of Pharmacy, Medical University of Warsaw, Banacha 1 Str., 02-097 Warsaw, Poland; s082570@student.wum.edu.pl (M.M.); s082651@student.wum.edu.pl (K.P.); magdalena.bujalska@wum.edu.pl (M.B.-Z.); 2Department of Medical Biology, Medical University of Bialystok, Mickiewicza 2c Str., 15-222 Bialystok, Poland; elzbieta.muszynska@umb.edu.pl; 3Department of Psychiatry, Faculty of Pharmacy, Medical University of Warsaw, Nowowiejska 27 Str., 00-665 Warsaw, Poland; tadeusz.nasierowski@wum.edu.pl

**Keywords:** magnetic resonance imaging, manganese, manganism

## Abstract

In recent years, a series of articles has been published concerning magnetic resonance imaging (MRI) studies in a group of patients exposed to manganism, specifically factory workers, welders, and individuals with liver diseases, as well as those abusing home-produced ephedrone. Some potential symptoms of manganese toxicity include motor disturbances, neurocognitive problems, sleep disorders, and psychosocial changes. Despite various publications on MRI research in individuals with an elevated risk of manganism, there is a noticeable absence of a comprehensive review in this field. The detection of the accumulation of manganese in the brain through MRI can confirm the diagnosis and guide appropriate treatment. Due to the high cost of determining manganese ion levels in biological material, an additional aim of the manuscript was to identify simple medical laboratory parameters that, when performed concurrently with MRI, could assist in the diagnosis of manganism. Among these types of parameters are the levels of bilirubin, magnesium, liver enzymes, creatinine, hemoglobin, and hematocrit.

## 1. Introduction

Manganese poisoning, also known as manganism, is a syndrome of symptoms that develop due to chronic exposure to manganese compounds. This medical condition was first described in 1837 by James Couper and colleagues [1]. Under normal circumstances, the human body requires small amounts of manganese for proper functioning, as it plays a crucial role in various biological processes such as metabolism, bone development, brain function, and hormonal balance [2,3]. However, excessive exposure to manganese, especially in occupational settings, can lead to the development of manganism. For manganese, Occupational Safety and Health Administration (OSHA) has set a permissible exposure limit (PEL) of 5 milligrams per cubic meter (mg/m^3^) of air, calculated as an 8 h time-weighted average (TWA) concentration. This means that workers should not be exposed to airborne concentrations of manganese exceeding this limit during an 8 h work shift [4]. The primary groups at risk of manganese exposure include factory workers, miners, welders, and individuals consuming homemade ephedrone [4,5,6]. The accumulation of manganese in tissues, particularly in the brain, can result in various neurological symptoms [7]. The most common symptoms of manganism include movement disorders, tremors, muscle stiffness, speech problems, and psychological challenges such as depression or concentration difficulties. Prolonged exposure to manganese can lead to permanent damage to the nervous system [8,9]. An advanced medical imaging technique used for diagnosis is magnetic resonance imaging (MRI), which utilizes magnetic fields and radio waves to obtain detailed images of internal body structures [10]. MRI can be a valuable tool in diagnosing manganism, as manganese has the ability to accumulate in specific brain regions, especially the substantia nigra, which is crucial for motor control [11]. In magnetic resonance imaging studies, a method called T1-weighted imaging allows the detection of manganese accumulation in tissues, including the substantia nigra. Such studies can provide information about the presence of manganese in the body, which is significant when there is the suspicion of manganese exposure [12,13]. Identifying manganese accumulation in the brain through MRI can help confirm the diagnosis and guide appropriate treatment. Despite numerous publications on MRI studies in patients particularly exposed to manganism, comprehensive reviews are lacking in this area. Therefore, the objective of this manuscript was to review articles related to MRI studies in the group of patients exposed to manganism, specifically among factory workers, miners, welders, patients with liver diseases, and individuals consuming homemade ephedrone. The second aim of the review was to identify simple medical laboratory tests that could assist in diagnosing manganism during MRI examinations. This recommendation stems from the high cost and the lack of widespread availability for measuring manganese ion concentrations in numerous medical laboratories worldwide. In this review, a comprehensive search was conducted on international databases including Thomson (Web of Knowledge), PubMed/Medline, Science Direct, Scopus and Google Scholar to identify all clinical trials, case reports, review articles, and meta-analyses related to MRI findings in manganese poisoning. The articles in the reference section of all studies taken into consideration were screened. Search terms included “manganism in MRI”, “manganese poisoning” and “manganese MRI”.

## 2. Factory Workers, Miners and Welders

Mn exposure and Mn poisoning pose significant concerns for miners, welders and other workers exposed to manganese, primarily due to its neurotoxic effects. Over the years, numerous studies have been conducted to investigate the impact of Mn exposure, the potential risk of developing manganism and its MRI markers. Neurological manifestations stemming from Mn exposure commonly manifest after many years of contact with this microelement. Thus, a thorough understanding of the potential risks inherent in the occupational activities undertaken is of paramount importance. Nevertheless, reports in the literature suggest that initial manifestations of Mn toxicity in welders may emerge as early as one month following exposure. This length of time depends on the duration and severity of exposure, the toxic forms involved (fumes or dust), the limitation of ventilation and intraindividual differences [14]. Furthermore, as indicated by certain researchers, neurotoxic effects are more prevalent among smelters compared to welders, irrespective of the duration and level of exposure [15]. The most prevalent neurological manifestations among Mn-exposed workers are extrapyramidal symptoms, Parkinsonism characterized by rigidity, bradykinesia, tremor and dystonia [16]. In rare cases, symptoms can manifest distinctly, such as myoclonic involuntary movements (IVM), occurring without the concurrent presence of Parkinsonism [17]. The neurotoxic mechanisms associated with manganese-induced Parkinsonism differ from those observed in Parkinson’s Disease (PD). However, prolonged welding duration and increased exposure levels may induce neurotoxic alterations in the substantia nigra, a pathological locus implicated in PD [18]. According to Martin et al. unlike Parkinson’s disease (PD), manganese-induced Parkinsonism may result from impaired (too low) dopamine release [19]. PET/SPECT are used to differentiate idiopathic Parkinson’s disease (IPD) from manganese-induced Parkinsonism (in the latter, PET is normal) [20]. The clinical syndrome, lack of response to levodopa [14,21], imaging studies with MRI and PET, and pathologic features all help to distinguish these two conditions in welders and permit the correct diagnosis to be established [22]. Researchers have demonstrated that manganese exposure, as assessed through personal monitoring, is significantly linked to deteriorated handwriting stability in welders. Symptoms including extrapyramidal, olfactory and mood disturbances did not improve over time and may have even worsened, while cognitive function appeared to improve in retired welders. A shorter T1 was associated with lower performance on verbal fluency, verbal learning, memory and preservation tests. The MRI image was pathognomonic for manganism but was similar to images seen, for example, in hypermagnesemia [23]. Currently, the signal intensity on a T1-weighted MRI for the globus pallidus, caudate and putamen and pallidal index is the most reliable MRI-related biomarker for the diagnosis of manganese toxicity [24]. The T1-weighted basal ganglia signals and pallidal index are associated with occupational Mn exposure and the severity of Parkinsonism in steelworkers, miners and welders [25,26,27]. However, these MRI abnormalities are not directly related to the symptoms of manganism, as they can also occur in asymptomatic welders [28]. Some researchers suggest that blood Mn level only correlates with T1 relaxation time, not with PI in welders [29]. R1 (1/T1) but not R2 (1/T2) might be an indicator of metal accumulation, particularly in the globus pallidus. Additionally, elevated airborne Mn concentration is linked to increased R1 signals in this brain region [30,31]. Kim Y et al. investigated the relationship between signal intensity on T1-weighted MRI in asymptomatic workers. The authors of that paper indicate that the increase in signal intensities on the T1-weighted image reflect recent exposure to Mn, but not necessarily manganism [32]. Many Mn-exposed workers who do not have any clinical symptoms may have hyperintensities in the brain MRI. Therefore, increased signal intensities on T1-weighted images may reflect exposure to Mn or an accumulation of Mn, but not necessarily manganism [33]. Other studies suggest that an increased-intensity T1 MRI signal emerges even in individuals with normal blood manganese levels, implying that this MRI finding primarily serves as a biomarker for chronic Mn intoxication. On the other hand, blood Mn concentration might only reflect short-term Mn exposure, with limited relevance to prolonged or past exposure [34,35,36]. Intensity indices are higher in Mn-exposed miners with the longest cumulative Mn exposure [37]. Moreover, Lee E. Y. et al. suggest that R1 signals of the basal ganglia might serve as a more sensitive marker for capturing short-term dynamic changes in Mn accumulation than the pallidal index in welders [38]. Some researchers suggest that when nothing is visible on MRI, it means that exposure to manganese ended 6–12 months ago [20]. Using MRI indicators, it is challenging to determine which parameter values indicate toxicity and which indicate elevated levels without toxic effects. The concentration of manganese in the blood does not reflect the current manganese content in the brain due to the long half-life of manganese in the brain. MRI is effective in qualitative diagnoses but less accurate in quantitative diagnoses [39]. Some researchers propose a connection between T1 and poorer outcomes in certain neuropsychological tests, yet substantial evidence confirming this hypothesis is lacking in Mn-exposed workers [40]. There is no direct association between fine motor test results and the relaxation rates R1 in globus pallidus and substantia nigra in welders [41]. Other researchers have used the pallidal index (PI) as a parameter to evaluate the Mn accumulation in the brain, which is calculated as the signal intensity ratio of GP relative to the frontal white matter (FWM) on T1-WI. Lee E. Y. et al. propose that Mn accumulation can be more sensitively evaluated using MRI R1 values compared to PI, particularly at lower exposure levels. The reduced sensitivity of PI as a marker is likely attributable to the accumulation of Mn in the orbitofrontal white matter in welders [42]. PI is considerably more useful for assessing the Mn exposure of asymptomatic Mn-exposed smelters who are still working, compared to evaluating blood manganese concentration [36]. Other MRI-related markers of manganism are also under examination. Long Z. et al. propose that short-echo-time magnetic resonance spectroscopy (MRS) of GABA levels in thalamus could potentially serve as a marker for subtle impairments in fine motor skills within the population occupationally exposed to manganese. The study by Long et al. revealed a notable association between fine motor abilities and GABA concentrations in the thalamus, implying that altered thalamic GABA status may at least partially underlie the mechanisms of motor deficits. Moreover, it appears that GABA levels in the thalamus increase with prolonged manganese exposure in smelters [43]. Chang Y. et al. suggest that volumetric measurements could serve as a valuable subclinical marker among asymptomatic welders. Researchers observed measurable reductions in brain volume within the globus pallidus and cerebellum of welders with chronic manganese exposure. These volume reductions were found to be associated with cognitive and motor neurobehavioral impairments in welders [44].

## 3. Patients with Liver Diseases

Patients after liver transplantation are vulnerable to manganese toxicity presenting as Parkinsonism–dementia complex with rapid onset [45]. Katsuragi et al. showed that cerebrospinal fluid (CSF) manganese concentration is correlated with pallidal hyperintensities on T1-weighted MRI, and therefore a useful marker for neurotoxicity induced by manganese poisoning; however, they did not conduct neurophysiological tests [46]. Homozygotous mutations in the SLC30A10 gene encoding manganese transporter have been correlated with manganese depositions in anterior pituitary, caudate, lentiform and dentate nuclei and cerebellar white matter. Tuschl K et al. showed that iron supplementation and chelate therapy improve clinical performance in such patients [47]. Pallidal hyperintensity on T1-weighted MRI scans often arises as a significant consequence of manganese deposition. In cases of chronic manganese intoxication unaccompanied by concurrent liver disease, this phenomenon also results in elevated signal intensity within the pallidal region on MRI scans [48,49]. Prolonged exposure to manganese, with increased intervals between doses and longer durations of exposure, triggers microglial activation and neuroinflammation, resulting in the depletion of dopaminergic neurons in the brain. Furthermore, it induces liver inflammation, causing liver damage and initiating the Nrf2 antioxidant pathway, along with the upregulation of efflux transporters as adaptive responses to chronic manganese toxicity [50]. The liver regulates the redistribution of excessive manganese to specific tissues or organs and facilitates its clearance through the hepatobiliary system. However, mutations in manganese transporters disrupt this homeostasis, leading to hepatic damage and an accumulation of excess manganese in the body. The liver plays a crucial role in excreting manganese, and any disruption to this process can lead to manganese-induced neurodegeneration in mammals. Disturbances in manganese homeostasis can detrimentally impact the integrity of the Golgi apparatus, thereby disrupting the normal trafficking of manganese in WIF-B cells [51,52]. Chronic liver failure can lead to manganese toxicity. According to Aissi M et al. [53], the occurrence of T1 hyperintensities in both pallidal regions on a brain MRI may suggest the presence of hypermanganism [54], even when observed in a clinical context marked by nonspecific neurological symptoms such as psychomotor impairment. Alcoholic impairment of the liver may also lead to Mn-induced Parkinsonism [55]. Moreover, it is essential to contemplate the possibility of manganese poisoning when confronted with dyskinesia and the distinctive brain MRI patterns. Even when there is no known environmental exposure, situations such as total parenteral nutrition or persistent hepatic dysfunction can give rise to significant manganese accumulation, leading to the emergence of symptomatic symptoms [56]. In the study by Ikeda S. et al. on the postoperative phase following biliary atresia treatment, certain patients exhibited manganese deposits in the globus pallidus on T1-weighted MRI images, accompanied by elevated manganese levels in their whole blood. This occurrence may be attributed to an augmented portosystemic shunt, and it is associated with the presence of a latent or subclinical encephalopathy [57]. Manganese deposits in T1-weighted MRI were also reported in patients with biliary atresia who had undergone Kasai’s portoenterostomy [58]. Progressive myelopathy is an uncommon occurrence linked to prolonged liver disorders. The patient described by Gospe SM Jr et al. had ongoing polycythemia and remarkable rises in total blood manganese levels, accompanied by magnetic resonance imaging confirming manganese accumulation in the basal ganglia and other brain regions. Notably, the patient did not develop liver failure or Parkinsonism during this period [59]. These results are consistent with the work of Wilhelm Nolte et al., which proved that bright basal ganglia are not directly associated with disturbed liver function, nor with portal-systemic encephalopathy [60].

## 4. Patients Taking Ephedrone

Ephedrone encephalopathy denotes a cluster of symptoms arising from the accumulation of manganese in the central nervous system (CNS), which arises due to the misuse of ephedrone (methcathinone). This substance is produced through a DIY process involving ephedrine and an excessive quantity of oxidant—potassium permanganate. Nervous system impairment can manifest after around six months’ use of ephedrone tainted with manganese. Nevertheless, it has been demonstrated that not every person utilizing homemade ephedrone will exhibit these symptoms [61]. Users of ephedrone experience the development of encephalopathy symptoms much more rapidly than welders or miners (within 5–9 months). Assessing cognitive function can serve as a sensitive indicator of manganese neurotoxicity dynamics, especially in less advanced (subclinical) cases. It is likely to be less useful in advanced stages of manganese poisoning, as cognitive impairments appear to be more persistent during that time [62]. In the study by Stepens A. et al., former users of phedrone i.v. experienced a notable decrease in manganese levels, and there was a reduction in T1-weighted hyperintensity observed in the basal ganglia MRI for three out of four former users and two out of three individuals who had discontinued use. It is worth noting that this improvement in imaging findings did not correspond to clinical improvement (Mn-induced movement disorder) [63]. Patients poisoned with manganese from home-made ephedrone present with cock-gait [62,64]. Presumably, changes in T1-weighted MRI decline after several years of abstinence [65,66]. However, ceasing the use of self-produced ephedrone has a more significant impact on reducing the severity of motor symptoms compared to cognitive symptoms in two years’ prospective study [62]. Ephedrone-induced manganism is characterized by having no response to levodopa [67,68]. One of the prodromal symptoms of manganese poisoning is slurred speech [69]. Dysarthria emerged after an average of 8.5 ± 3.2 months of consistent intravenous ephedrone misuse, serving as the initial symptom in one-third of cases. A differential diagnosis of chronic manganese toxicity with Parkinsonism may take into account perceptual speech analysis [64]. Hypophonia and gait disturbance were also considered to be the first sign of manganism in the study by Stepens A. et al.; however, the onset was 5.8 years [70]. T1-weighted MRI images show that brain regions mostly affected by an excess of manganese include symmetric bilateral hyperintensity in the globus pallidus [71], dentate nucleus, subcortical white substance of the cerebellar hemisphere, and putamen [68]. However, Jurmaa et al. have shown subcortical grey matter atrophy in methcathinone abusers within the putamen and thalamus bilaterally, and the left caudate nucleus. Also, in this study, there was a negative correlation between the volume of the caudate nuclei and the duration of abuse [72,73]. The use of positron emission tomography with 6-[18F]-fluorodopa indicated a minor reduction in pre-synaptic uptake, specifically in the back portion of the striatum. This suggests that the dopaminergic nigrostriatal system has been somewhat protected from the pathological process [74].

## 5. Genetic Susceptibility and Environmental Triggers

Inherited manganism may present with severe, dystonic dysarthria; asymmetrical, generalized dystonia, worse in the right inferior limb; bilateral, striatal toe; tip-toe or cock-walk gait; bilateral, sensorineural hearing loss; urinary incontinence; and constipation [75]. These symptoms have been linked to SLC30A10 gene mutations [76]. In the cases of SLC30A10 gene mutations, neuroimaging has shown numerous regions with hyperintensities on T1-weighted images, encompassing the basal ganglia and dorsal brainstem. Therefore, the mutation in the SLC39A14 gene (Mn transporter) has a similar presentation in MRI. In this case, administering intravenous CaNa2EDTA resulted in a noteworthy decrease in both serum manganese levels and T1 hyperintensities. However, the improvement in dystonia was marginal. Consequently, the early detection of this genetic disorder is imperative, given its potential for treatment [77]. The cause of manganism may be related to diet. Typical symptoms include polycythemia, characteristic neurological features (“cock-walk gait”), and bilateral hyperintensity in the basal ganglia on T1-weighted MRI [78]. Sahni et al. demonstrate that, despite elevated manganese levels in the blood, symptoms may not occur, and there is no strict correlation between blood manganese levels and the degree of toxicity. For instance, the patient’s 7-year-old sibling was asymptomatic with almost identical exposures and elevated plasma Mn levels, while a previously healthy 6-year-old child presented with severe Mn neurotoxicity. An exposure assessment revealed seasonal ingestion exposure to Mn at the family’s summer cottage, common to all four immediate family members. Well water used for drinking and cooking exceeded recommended guidelines, and their diet was characterized by foods high in Mn [79]. In the literature, there are descriptions of cases where patients developed manganism due to the consumption of, for example, large quantities of black tea infusions [80], chemically contaminated water [81], or gasoline in a suicide attempt [82].

In Figure 1, we have illustrated the exposed population, risk factors, potential toxicities of manganese, and MRI findings. A summary of the most important research on MRI findings and the clinical manifestation of manganism in various populations is shown in Table 1. 

## 6. Discussion and Laboratory Recommendations

The conducted review is the first, to the best of our knowledge, addressing the association between MRI and manganism in specific groups of individuals. The articles cited in this manuscript suggest that imaging studies such as MRI can be utilized to assess potential brain damage related to excessive manganese exposure. However, these findings are not unequivocal. Some studies suggest a correlation between manganese levels and parameters associated with MRI, while others indicate the difficulty in determining which values of these parameters allow for the assessment of manganism exposure. Additionally, blood manganese levels may not serve as an adequate indicator of the current amount of this element in the brain, given the prolonged retention time of this trace element in that part of the body. Therefore, it is recommended to measure simpler laboratory parameters to complement the results of MRI examinations in patients exposed to manganism. The diagnosis of manganism involves various methods, including laboratory tests. Measuring manganese concentration in the blood may be one of the diagnostic criteria. However, the high cost of determining manganese ion levels in biological material, coupled with a lack of available technical equipment, justifies the search for alternative biomarkers for manganism [83]. 

The first recommendation involves measuring the level of bilirubin in the serum. Several published studies indicate that patients with hyperbilirubinemia exhibit visually apparent typical imaging symptoms in existing traditional MRI sequences [84]. Stamelou et al. pointed out that a patient with a confirmed homozygous mutation in SLC30A10 had mild hyperbilirubinemia accompanied by an elevated manganese level in whole blood [85]. Another account pertains to a patient experiencing recurrent neuropsychiatric illness resulting from chronic manganese toxicity linked with antibodies targeting voltage-gated potassium channels (VGKC). In this patient, elevated unconjugated bilirubin levels (35 µmol/L, norm < 25) were diagnosed, and liver and bile duct ultrasonography revealed an enlarged gallbladder with sediment content and possibly some thickening of the wall [86]. According to one experimental model, the combination of manganese with bilirubin may lead to an increase in intracellular cholesterol levels, adversely affecting bile formation and flow processes [87]. 

The next recommendation involves measuring the levels of liver enzymes, specifically alanine aminotransferase and aspartate aminotransferase. Available data suggest that liver function values from MRI images are significantly associated with overall liver function parameters, allowing simultaneous information about both function and anatomy using a single research technique [88]. Manganese exposure leads to the inhibition of antioxidant enzymes, such as superoxide dismutase (SOD) and glutathione peroxidase (GPx). Furthermore, a decrease in glutathione (GSH) levels and an increase in malondialdehyde (MDA) levels in liver tissues have been observed. Manganese also shows inhibitory effects on SOD activity, leading to an increase in MDA levels in hepatocyte nuclei. Additionally, a reduction in the activity of the sodium–potassium pump in hepatocytes has been observed, associated with elevated MDA levels and decreased cell membrane fluidity [89]. The results published in 2023 in the European Journal of Medical Research indicated a positive association between blood manganese levels and liver stiffness in a group of 4690 individuals with chronic obstructive pulmonary disease (COPD). Elevated blood manganese levels are associated with more severe liver damage [90]. This is linked to the fact that the liver plays a crucial role in the accumulation and metabolism of manganese in the body [91]. 

The next recommendation involves measuring the level of creatinine. Studies conducted on 240 MRI scans of patients with chronic kidney disease showed that biomarkers such as estimated glomerular filtration rate (eGFR) and urine albumin–creatinine ratio (UACR) were correlated with observed changes in brain structure in magnetic resonance imaging (MRI) studies, even after considering vascular risk factors [92]. In 2020, in the Journal of Renal and Hepatic Disorders, authors noted that precise mechanisms of manganese nephrotoxicity had not been identified. The study showed that administering manganese for 30 consecutive days was associated with a significant increase in blood urea nitrogen (BUN) and serum creatinine. Additionally, an increase in glucose, phosphate, and protein levels in urine was observed, while histopathological changes included the atrophy of renal tubules, interstitial inflammation, and necrosis [93]. Manganese intake may contribute to a significant increase in reactive oxygen species (ROS) production and affect the levels of oxidative-stress-related biomarkers in kidney tissue [94]. 

The subsequent recommendation involves measuring hemoglobin and hematocrit levels. Yildirim et al. indicated that MRI examination was a valuable tool in a group of patients with chronic anemia. The results showed that a decreased intensity of cranial bone marrow (CBM) may indicate anemia, and increased CBM thickness may specifically indicate hemolytic anemia [95]. Circulating manganese in the body is usually bound to hemoglobin. Therefore, erythrocytes are the primary compartment for manganese [96]. Considering the similarities between manganese and iron, the regulation of both metal balances is interconnected. Hence, the iron status affects manganese accumulation. An example of this is the situation in anemia, where low iron levels promote manganese absorption [97]. There are clinically relevant interactions between manganese and iron [98]. Manganese levels are higher in patients with low ferritin levels, both in women and men [99]. Serum ferritin level is the main determinant of Mn levels in blood [100]. Increased manganese content in biological samples from welders is associated with decreased iron content in serum and erythrocytes [101]. 

The final recommendation involves measuring magnesium ions. The study, published in the European Journal of Nutrition, conducted on a group of 6001 participants, examined the association between dietary magnesium intake and brain volume and white matter lesions (WML) in middle and early older age. Higher dietary magnesium content was associated with greater brain volume and lower WML [102]. In the discussion of an article on manganese ion levels in patients abusing ephedrone, the authors highlight the justification for measuring this macroelement [83]. Due to the similarity in physicochemical properties between magnesium and manganese, most enzymes can activate using magnesium instead of manganese [103]. Studies conducted on animal models have shown that insufficient magnesium intake in the diet can affect manganese metabolism [104]. Magnesium supplementation at a dose of 200 mg per day results in a reduction in manganese bioavailability, probably by reducing manganese absorption or increasing its elimination [105]. 

In summary, the recommended simple laboratory parameters should be assessed simultaneously in patients exposed to manganism undergoing MRI examinations. In the future, it would be valuable to analyze these parameters to determine their correlation with manganese levels. Among the main parameters of this kind are bilirubin level, liver enzymes, hemoglobin, and serum creatinine (Table 2). These are simple laboratory parameters, and it should be noted that not all laboratories have the capability to measure the level of manganese. Greater emphasis should be placed on the search for additional biomarkers, used in conjunction with MRI, for the diagnosis of manganism. 

## Figures and Tables

**Figure 1 jcm-13-02823-f001:**
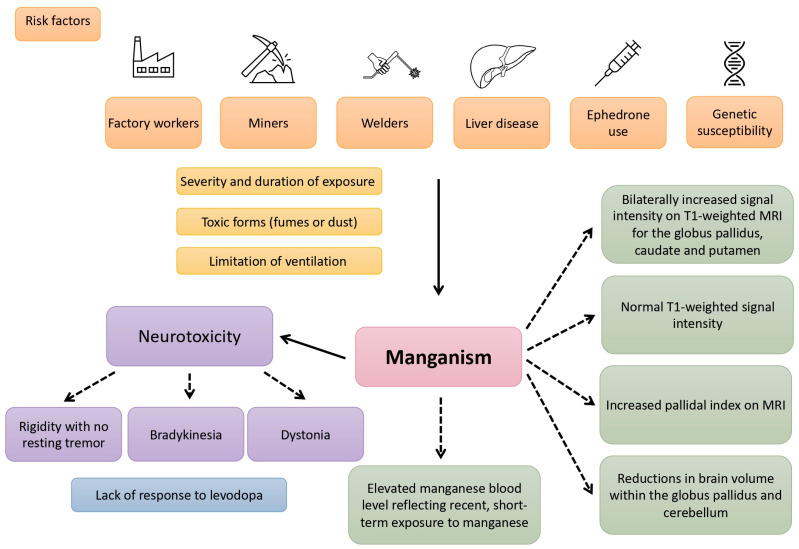
A picture depicting the exposed population, risk factors, potential toxicities of manganese, and MRI findings.

**Table 1 jcm-13-02823-t001:** The most important research on MRI findings and clinical manifestation of manganism in various populations.

Group of Patients	Participants in Study	Study Description	Results	Reference
Factory workers	Study group: 121 factory workers	Conducted: manganese blood level, MRI of the brain.	The signal index on T1-weighted MRI demonstrated a dose–response relationship. Signal intensity also serves as an effective predictor of neurobehavioral performance in manganese-exposed workers. However, even individuals with low manganese exposure exhibited significantly increased MRI intensity in the basal ganglia—no threshold value was found at which MRI did not show these abnormalities.	[28]
Factory workers	Study group: 121 male factory workers randomly selected out of a total of 750 workers, including Mn-exposed, non-exposed manual, and non-exposed clerical workers	Conducted: MRI of the brain, neurological examination.	Increase in signal intensities on the T1-weighted image reflect recent exposure to manganese but not necessarily manganism.	[32]
Miners	Study group: 19 manganese-exposed miners and 10 other miners	Conducted: MRI of the ex vivo brain tissue.	Typical elevation basal ganglia T1 indices when compared to controls. Predictors of ex vivo T1 MRI signal intensity in manganese-exposed mine workers include duration of manganese exposure and neuronal density in the putamen (*p* = 0.05) and caudate (*p* = 0.04)	[37]
Welders	Study group: 43 welders exposed to manganese and 31 healthy controls	Conducted: MRI of the brain.	Diffusion tensor imaging fractional anisotropy (estimate of microstructural integrity) values in the substantia nigra and globus pallidus correlate with cumulative lifetime welding exposure (*p* < 0.03). Welding exposures above a certain level may induce neurotoxicity in the substantia nigra.	[18]
Welders	Study group: 111 male welders exposed to manganese	Conducted: manganese blood level, neuropsychological tests.	Pallidal index is the most effective predictor of neurobehavioral performance after adjusting for age and level of education.	[24]
Welders	Study group: 20 male welders and 10 non-office-workers in a shipyard	Conducted: manganese blood level, MRI of the brain.	The blood manganese level showed non-linear correlation solely with T1 relaxation time, not with pallidal index. Both T1 and pallidal indices serve as effective biomarkers for environmental manganese exposure.	[29]
Welders	Study group: 40 welders and 26 healthy controls	Conducted: manganese blood level, MRI of the brain (including measurement of the volume of the globus pallidus and cerebellum).	Brain volumes in the globus pallidus and cerebellar regions were significantly diminished in welders with chronic manganese exposure compared to controls (*p* < 0.05).	[44]
Patients with liver diseases	Study group: 5 participants (including 2 patients with cirrhosis) and 10 healthy controls	Conducted: manganese blood and cerebrospinal fluid (CSF) levels, CT and MRI of the brain.	T1-weighted MRI hyperintensity correlates with the CSF manganese concentration. CSF manganese concentration may be a most useful marker and predictor of manganese neurotoxicity in patients with symmetric pallidal hyperintensities on T1-weighted MRI.	[46]
Patients with liver diseases	Study group: 4 patients who underwent Kasai’s portoenterostomy	Conducted: manganese blood level, MRI of the brain.	Patients with biliary atresia who have undergone Kasai’s portoenterostomy have a risk of latent manganese toxicity. MRI scans of the brain and manganese levels in the blood can be used for early detection of this complication.	[58]
Patients with liver diseases	Study group: 6 patients with complete portal vein thrombosis but without chronic liver disease	Conducted: manganese and ammonia blood levels, MRI of the brain, EEG, neuropsychological tests.	Hyperintensities on T1-weighted MRI of basal ganglia are not directly associated with disturbed liver function nor with portal-systemic encephalopathy.	[60]
Patients taking ephedrone	Case report: a 19-year-old male home-made-ephedrone abuser with HIV and HCV infection	Clinical picture: elevated manganese blood level, behavioral disorder, memory loss, psychomotor retardation, drowsiness, speech disorder, gait and postural balance disorder persisting for several months. The patient synthesized ephedrone at home from pseudoephedrine-containing drugs using potassium permanganate.	An adverse effect of home-made-ephedrone synthesis might be a high load of manganese contamination, which may lead to manganism and brain damage.	[61]
Patients taking ephedrone	Study group: 23 adult abusers of home-made-ephedrone who had extrapyramidal symptoms	Clinical picture: elevated manganese blood level; 23 patients had gait disturbances and difficulty walking backward, and one of them used a wheelchair; 21 patients had hypophonic speech, and one of these patients was mute. None of the patients reported worsening cognitive function. Hyperintense signals in the basal ganglia bilaterally in the T1-weighted sequence on MRI. The patients synthesized ephedrone at home from pseudoephedrine-containing drugs using potassium permanganate.	Manganese present in the methcathinone solution leads to a persistent neurological disorder. Among those who ceased using ephedrone, none reported significant improvement in gait or speech. In contrast to Parkinson’s disease patients, these individuals did not exhibit resting tremors.	[70]
Patients taking ephedrone	Case report: a 30 years old male home-made-ephedrone abuser	Clinical picture: elevated manganese blood level, two-year history of gait disturbance, hypophonia, hypomimia, mild bradykinesia and rigidity with no resting tremor, hyperintense signals in the basal ganglia bilaterally in the T1-weighted sequence on MRI. The patient synthesized ephedrone at home from pseudoephedrine-containing drugs using potassium permanganate.	Manganism secondary to ephedrone abuse causing Parkinsonism should be suspected in cases of atypical Parkinsonism. Patients with manganese-induced Parkinsonism do not respond to levodopa treatment.	[71]
Patients with inherited manganism	Case report: a 19-year-old female with manganism and (c.922C>T [p.Gln308*]) mutation of the SLC30A10 gene	Clinical picture: elevated manganese blood level, dystonia, dysarthria, polycythemia, hyperintense signals in the basal ganglia bilaterally in the T1-weighted sequence with normal T2-weighted sequence on MRI; percutaneous liver biopsy revealed mild cirrhosis with negative staining for iron, copper, and manganese.	The genetic etiology of manganism and the investigation of SLC30A10 mutations should be considered in idiopathic cases of manganism.	[75]
Patients with inherited manganism	Case report: a 18-year-old female with manganism and (c.367C>T, p.Q123*) and (c.512G>A, p.G171E) mutation of the SLC39A14 gene	Clinical picture: elevated manganese blood level, wheelchair user, generalized dystonia, iron-deficiency anemia, no history of dystonia in her family and no environmental exposure to heavy metals, hyperintense signals in the basal ganglia bilaterally in the T1-weighted sequence on MRI.	A diagnosis of inherited manganism should be considered in idiopathic cases of manganism because it is potentially treatable. Early treatment with chelation with edetate calcium disodium (CaNa2EDTA) may prevent disease progression and even reverse the established deficits.	[77]

**Table 2 jcm-13-02823-t002:** Recommended laboratory parameters to detect manganism.

Recommended Laboratory Parameters to Detect Manganism	Normal Range	References
Bilirubin in serum	<1 mg/dL	[85,86,87,106]
Liver enzymes (alanine aminotransferase and aspartate aminotransferase) in serum	AST < 35 IU/L ALT < 55 IU/L	[88,89,90,91,107]
Creatinine	0.6–1.5 mg/dL	[92,93,94,108]
Hemoglobin and hematocrit level	Haemoglobin: 12–16 for woman; 13–18 for men Hematocrit: 40–54% for men; 36–48% for woman	[95,96,109,110]
Iron in serum	11–33 μmol/L	[97,98,99,100,101,111]
Magnesium in serum	0.1.6–2.4 mg/dL	[83,101,102,103,104,105,112]

## Data Availability

Not applicable.

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
