# Peer review of "Magnetic Resonance Imaging and Manganism: A Narrative Review and Laboratory Recommendations"

_jcm, 2024, doi:10.3390/jcm13102823_

Round 1

Reviewer 1 Report

Comments and Suggestions for Authors

The review is focussing on Mn toxicity and laboratory recommendations to detect manganism. There are some suggestions to the authors.

1. Authors should summarize the studies in table format as it makes easier to compare all the studies at same time

2. Author can include a picture depicting the exposed population and potential toxicities of Mn and risk factors

3. Again, a table could be useful to highlight the recommended laboratory methods to detect manganism including the normal range of, for example, bilirubin etc

4. If the miners, welders etc are exposed to Mn, is there any tolerance range?

Comments on the Quality of English Language

English language is acceptable

Author Response

Dear Reviewer nr 1, 

Comment 1: The review is focussing on Mn toxicity and laboratory recommendations to detect manganism. There are some suggestions to the authors.

Answer 1: Thank you very much for sending your comments related to our manuscript, as well as the associated positive word.

Comment 2: Authors should summarize the studies in table format as it makes easier to compare all the studies at same time.

Answer 2: The summary table has been inserted.

Comment 3: Author can include a picture depicting the exposed population and potential toxicities of Mn and risk factors.

Answer 3: The picture indicated by the reviewer has been added to the manuscript.

Comment 4: Again, a table could be useful to highlight the recommended laboratory methods to detect manganism including the normal range of, for example, bilirubin etc.

Answer 4: In accordance with the advice received from the reviewer, a table of this type was inserted into the manuscript. The table contains basic laboratory parameters aiding in the detection of manganese, including the normal range and references.

Comment 5: If the miners, welders etc are exposed to Mn, is there any tolerance range?

Answer 5: In the introduction, based on the advice received, an answer to this question was provided.

Reviewer 2 Report

Comments and Suggestions for Authors

The manuscript is a literature review on the diagnosis of manganese poisoning. The review is interesting, but I have some observations about the text.

Lines 59-61: I understand that the work is a review, and not a systematic review or meta-analysis, but this text could include literature search strategies.

Lines 82-84: The sentence begins and ends with "according to"

Lines 163-149: This excerpt from the manuscript is a very long paragraph, difficult to read. Some information could be included in a table to make it easier for readers to understand.

Lines 140-141: The present study?

Lines 151-187: The authors could briefly explain the pathophysiology of manganese accumulation in cases of liver disease.

Line 213: [64] Differential - include the point

Line 243: toxicity For instance

Lines 255-263: A table summarizing the findings of the different articles can help in understanding and interpreting the data.

Lines 270-343: It is a very long paragraph, difficult to read. Separate groups of information into different paragraphs to make it easier to read. Several sentences do not refer to diagnosis, such as line 282 (and several others).

Lines 275-277: The way this sentence is written has no connection with the previous sentence.

The general conclusion of the manuscript is superficial, without indicating which parameters could actually be evaluated. The authors should reformulate the conclusions and separate them from the rest of the manuscript.

Author Response

Dear Reviewer nr 2,

Comment 1: “The manuscript is a literature review on the diagnosis of manganese poisoning. The review is interesting, but I have some observations about the text.”

Answer 1: We thank you for the valuable advice regarding the manuscript we've written.

Comment 2:Lines 59-61: I understand that the work is a review, and not a systematic review or meta-analysis, but this text could include literature search strategies.”

Answer 2: At the end of the introduction, several sentences were added to the article regarding strategies for finding articles.

Comment 3: „Lines 82-84: The sentence begins and ends with "according to"

Answer 3: We appreciate your pointing out the repetition; it has been removed.

Comment 4: “Lines 163-149: This excerpt from the manuscript is a very long paragraph, difficult to read. Some information could be included in a table to make it easier for readers to understand.”

Answer 4: In accordance with your advice and feedback received from the second reviewer, additional tables have been created and included in the manuscript. Additionally, the longest sentence has been corrected.

Comment 5: “Lines 140-141: The present study?”

Answer 5: The indicated sentence has been corrected.

Comment 6: “Lines 151-187: The authors could briefly explain the pathophysiology of manganese accumulation in cases of liver disease.”

Answer 6: Sentences regarding the pathophysiology of manganese accumulation in cases of liver disease have been added to the manuscript.

Comment 7: “Line 213: [64] Differential - include the point”

Answer 7: The indicated sentence has been corrected.

Comment 8: “Line 243: toxicity For instance”

Answer 8: Thank you very much for pointing out this minor error; it has been corrected.

Comment 9: “Lines 255-263: A table summarizing the findings of the different articles can help in understanding and interpreting the data.”

Answer 9: Thank you for another valuable piece of advice. Following it, along with the suggestion from the second reviewer, a table of this kind has been created.

Comment 10: “Lines 270-343: It is a very long paragraph, difficult to read. Separate groups of information into different paragraphs to make it easier to read. Several sentences do not refer to diagnosis, such as line 282 (and several others).”

Answer 10: The described recommendations have been separated. Each of them begins a new paragraph. The indicated sentence has been removed.

Comment 11: “Lines 275-277: The way this sentence is written has no connection with the previous sentence.”

Answer 11: This sentence pertains to the case description referenced in number 86. However, it has been corrected, meaning it has been rewritten differently.

Comment 12: “The general conclusion of the manuscript is superficial, without indicating which parameters could actually be evaluated. The authors should reformulate the conclusions and separate them from the rest of the manuscript.”

Answer 12: Following the earlier advice of the reviewer, this section has been divided into separate paragraphs. This also applies to the conclusions, which have been expanded with additional sentences.
